# Analysis of Intracellular Magnesium and Mineral Depositions during Osteogenic Commitment of 3D Cultured Saos2 Cells

**DOI:** 10.3390/ijms21072368

**Published:** 2020-03-30

**Authors:** Giovanna Picone, Concettina Cappadone, Alice Pasini, Joseph Lovecchio, Marilisa Cortesi, Giovanna Farruggia, Marco Lombardo, Alessandra Gianoncelli, Lucia Mancini, Menk Ralf H., Sandro Donato, Emanuele Giordano, Emil Malucelli, Stefano Iotti

**Affiliations:** 1Department of Pharmacy and Biotechnology, University of Bologna, 33-40126 Bologna, Italy; giovanna.picone2@unibo.it (G.P.);; 2Department of Electrical, Electronic and Information Engineering “Guglielmo Marconi” (DEI), University of Bologna 50, 47522 Cesena, Italy; 3Department of Chemistry “G. Ciamician”, Alma Mater Studiorum–Università di Bologna, via Selmi 2, I-40126 Bologna, Italy; 4Elettra-Sincrotrone Trieste S.C.p.A., Trieste, 34149 Basovizza, Italy; 5INFN section of Trieste, 2-34127 Trieste, Italy; 6Department of Medical Imaging, University of Saskatchewan, Saskatoon, SK S7N 5A2, Canada; 7Department of Physics, University of Calabria, 87036 Arcavacata di Rende (CS), Italy; 8National Institute of Biostructures and Biosystems (NIBB), 00136 Rome, Italy

**Keywords:** osteogenesis, osteoblastic differentiation, osteosarcoma, biomineralization

## Abstract

In this study, we explore the behaviour of intracellular magnesium during bone phenotype modulation in a 3D cell model built to mimic osteogenesis. In addition, we measured the amount of magnesium in the mineral depositions generated during osteogenic induction. A two-fold increase of intracellular magnesium content was found, both at three and seven days from the induction of differentiation. By X-ray microscopy, we characterized the morphology and chemical composition of the mineral depositions secreted by 3D cultured differentiated cells finding a marked co-localization of Mg with P at seven days of differentiation. This is the first experimental evidence on the presence of Mg in the mineral depositions generated during biomineralization, suggesting that Mg incorporation occurs during the bone forming process. In conclusion, this study on the one hand attests to an evident involvement of Mg in the process of cell differentiation, and, on the other hand, indicates that its multifaceted role needs further investigation.

## 1. Introduction

Accounting for 95% of its total, collagen is the major component of the organic matrix of bone tissue, the remaining 5% being proteoglycans and other non-collagenous proteins crucial in bone metabolism, such as osteocalcin, osteonectin, and osteopontin [1]. The mineral component of bone tissue is mainly represented by hydroxyapatite (HA), whose composition is prevalently calcium and phosphate, together with a small fraction of carbonates and other cation species [2]. Among the latter, magnesium plays an important role. Magnesium is the second most abundant intracellular cation and is involved in the activity of about 300 enzymes thus taking part in many aspects of bodily metabolism, such as muscle contraction, regulation of blood pressure, and bone cell function. Indeed, about 50% of the body’s magnesium can be found in bone tissue, mainly distributed in hydroxyapatite crystals, where it is understood to form a fixed and dynamic pool [3]. Its levels influence osteocytes proliferation and tissue remodelling. Interacting with hormonal regulators of bone metabolism, magnesium supports functioning of osteoblasts [4]. When the early stages of bone mineralization are considered, it has been demonstrated that magnesium increases the nucleation kinetic of HA downstream of expression of differentiation markers [5]. Although Mg has been shown to be involved in many cellular processes [6], the relationship between Mg and cell differentiation remains unclear [7].

Some studies indicate that low levels of extracellular magnesium promote cell differentiation, while others reach opposite conclusions. Fundamental differences exist between various cell types in the contribution of Mg to cell differentiation. Mg restriction retards cell differentiation of U937 cells towards osteoclasts, while mesenchymal cells differentiation toward pre-adipocytes seems to be independent from extracellular Mg [8]. Furthermore, Mg deficiency reversibly antagonizes the differentiation of promyelocytic leukemia HL60 cells exposed to differentiating agents such as DMSO and retinoic acid [9]. According to a previous report showing that high extracellular Mg inhibited matrix mineralization in prechondrogenic cell lines [10], it was recently shown that excessive extracellular Mg blocks pre-osteoblast differentiation [11].

Thus, in this work, we aimed at investigating magnesium content in a three-dimensional (3D) cell model during bone phenotype modulation. SaOS2 cells, a human osteosarcoma cell line, were selected for this investigation as they are credited for being a suitable in vitro model for studying the transition of human osteoblasts to osteocytes [12,13,14]. In addition, an added value of our investigation regards a potential for devising new antineoplastic strategies based on reversing undifferentiated cancer cells toward a less aggressive phenotype via molecular reprogramming [15].

In order to better mimic bone tissue structure and to improve the stability and the functionality of culture conditions, the cells were seeded within a 3D collagen scaffold. In fact, even though 2D systems allow to study the ability of tumour cells growth, they do not adequately take into account the physiological environment, failing to provide information about cell–cell and cell-extracellular matrix (ECM) interactions [16,17]. SaOS2 cells were induced to differentiate on a board of collagen scaffolds, collagen being the most abundant structural protein in bone tissue, widely employed in regenerative medicine [18]. Osteogenic phenotype of SaOS2 cells was monitored via RT-qPCR analysis of credited osteogenic markers (*RUNX2*, *COL1A1*, *BGLAP*, *SPP1*) and histological analysis of extracellular calcium depositions. Magnesium intracellular concentration was assessed by a fluorimetric method [19], while the presence and the distribution of the cation in mineral depositions was investigated by X-ray fluorescence microscopy [20,21]. Finally, a morphological characterization of samples was obtained by X-ray computed micro-tomography (µCT) techniques allowing to observe, by a non-destructive method, the distribution of extracellular mineral depositions, and their shape and size within the sample volume.

## 2. Results

### 2.1. SaOS2 Cells Growth on a Collagen Scaffold

SaOS2 cells were grown on board of collagen scaffolds and counted at 1, 3, 7, and 14 days to monitor cell proliferation (Figure 1). The doubling time was around 36 hours, in agreement with the literature regarding SaOS2 cell line during exponential phase growth. The viability of 3D cultured cells was about 80% until the 7th day of culture, and significantly decreased in the following days up to 60%, when the plateau phase was clearly visible (Appendix A). To better characterize the cell proliferative activity, DNA profiles were performed. Cell cycle analysis confirmed that cells were highly proliferating at day 3, while they slow down when reaching confluence: the percentage in G0/G1 phase went from 61% to 76%, and S phase halves from 29% to 15%. As such, the growth pattern in a 3D matrix indicates that a collagen scaffold is suitable for tissue-like growth of SaOS2 osteosarcoma cells in vitro.

### 2.2. Relationship between Osteogenic Differentiation and Magnesium Content of 3D Cultured SaOS2 Cells

SaOS2 cells were cultured on board of 3D collagen scaffolds for three and seven days, supplying fresh osteogenic cocktail twice a week. Cell osteogenic phenotype was assessed by both gene expression analysis of osteogenic markers and histological staining of extracellular calcium depositions. To clarify a potential role of magnesium in the differentiation process, the intracellular concentration of the cation was measured by means of the highly sensitive and performing DCHQ5 probe.

#### 2.2.1. Gene Expression Analysis after Osteogenic Treatment on SaOS2 Cells

Gene expression analysis of four osteogenic markers was evaluated by real-time qPCR, whose results are reported in Figure 2. Among the tested genes, the late osteogenic markers osteocalcin and osteopontin showed a strong increase of their mRNA expression levels, already after three days of treatment, compared to their controls. In detail, the upregulation of *BGLAP* and *SPP1* genes was 41.7-fold (*p*-value < 0.01) and 3.1-fold (*p*-value < 0.05) higher, respectively. After seven days of treatment, their levels were maintained overexpressed 47.4-fold and 5.5-fold, respectively, compared to their specific control counterparts (*p*-values < 0.01).

The early osteogenic commitment transcription factor *RUNX2* showed comparable level of expression during time, regardless of the osteogenic cocktail administered. On the other hand, although not significant, a trend in the upregulation of the other early osteogenic marker collagen type 1 (*COL1A1*) was scored, showing about a two-fold increase in its mRNA expression level at both time points with respect to control. Altogether, this gene expression profile suggests an osteogenic commitment of SaOS2 cells.

#### 2.2.2. Histological Analysis of Extracellular Matrix Calcium Depositions of Differentiated SaOS2

In order to assess the distribution of cells inside the collagen scaffolds, 10 μ-thick sections were marked by classical Hematoxylin/Eosin (H/E) staining. Figure 3a shows uniformly fitted cells.

Larger numbers of cells were found in control samples since a slowdown of proliferation is expected consistently with the commitment towards a more differentiated phenotype. Alizarin Red S staining was performed to evaluate calcium depositions usually associated with osteogenic commitment. Many Alizarin Red S positive nodular aggregates were present in treated samples, after only three days. The red spots were larger and more intensely stained at day 7, suggesting that a more extensive calcium deposition had occurred (Figure 3a). In addition, to quantify the size of the red (Alizarina Red staining) area covering the scaffold slices, four masks (*n* = 4) for each image acquired were analysed using the maximum entropy threshold-based image segmentation method. Figure 3 panel (c) shows a significant increase of areas covered by Ca depositions at both three and seven days after osteogenic induction.

#### 2.2.3. Monitoring of Magnesium Content during Osteogenic Differentiation of SaOS2 Cells

Intracellular magnesium content was measured by means of DCHQ5 fluorescent dye in differentiated cells. The treatment of 3D cultured cells determined a marked increase of magnesium concentration, at both three and seven days. In particular, a two-fold increase (arbitrary units) in the intracellular content of the treated samples with respect to their control has been observed (Figure 3b).

### 2.3. Depositions Characterization by X-ray Microtomography Techniques

With the aim to characterize and quantify the extracellular depositions, fixed scaffolds were imaged by phase-contrast microfocus X-ray µCT at the TomoLab station [22] of the Elettra synchrotron light source (Basovizza, Trieste, Italy). Figure 4 shows the 3D spatial distribution of the depositions in the whole scaffolds after seven days of osteogenic induction (panel (b)) and in a control sample (panel (a)). Two representative volumes of interest (VOIs) (panel (c) and (d)) inside the scaffold (CTRL volume = 1.8 mm^3^; treated volume = 1.2 mm^3^) were analysed by the ImageJ plug-in BoneJ [23] resulting in a higher amount of depositions fraction in seven days differentiated cells (0.80%) compared to the controls (0.31%) (table in Figure 4). This result suggests an increased biomineralization in differentiating SaOS2 cells. In addition, single-particles analysis showed that the volume of the depositions in differentiated cells was significantly lower (*p*-value < 0.001) with respect to the depositions present in the control samples (Figure 4). The graph classified the particles dimension (µm^3^) in relation to the number of particles within the class.

Appendix A of the scaffolds (CTRL Appendix A and treated Appendix A) highlighting the whole depositions distribution are reported.

Furthermore, a treated scaffold was imaged at the SYRMEP beamline of Elettra [24] by phase-contrast synchrotron X-ray µCΤ (Figure 5) reaching a spatial resolution of the order of 1 µm. Figure 5 panel (a) shows a reconstructed tomographic slice with a somewhat reduced spatial resolution that highlights the presence of mineral depositions along brighter filament indicating a not random pattern. Depicted in panel (b) of Figure 5 is the region of interest indicated in green in panel (a) with the full spatial resolution of the system. To better understand the spatial distribution of both depositions and filaments a 3D rendering of 50 adjacent slices is reported in Figure 5c.

A 3D rendering of depositions in a specific volume of interest is reported in Appendix A.

Depositions in differentiating 3D cultured SaOS2 cells were chemically characterized by X-ray fluorescence microscopy at the TwinMic beamline @ Elettra synchrotron [25]. Figure 4 panel (e) represents the elemental map of a deposition highlighting the co-localization of Mg and P pivotal elements of the HA.

## 3. Discussion

The involvement of magnesium in osteogenic differentiation is rather puzzling, mainly due to its complex and multifactorial role in cell metabolism. More and more evidences suggest that magnesium acts primarily as a signaling element in cell metabolism and the concept that Mg is an electrolyte is too simplistic and obsolete [8,26,27,28,29,30].

Recently, we reported that high extra-cellular Mg levels potentiate osteoclastic differentiation, while decrease osteoblastogenesis. A plausible hypothesis is that Mg might reprogram cholecalciferol (vitamin D3) activity on bone remodelling, causing an unbalanced activation of osteoclasts and osteoblasts [19]. On the other hand, we also demonstrated that magnesium deprivation accelerates the osteogenic differentiation of human bone marrow-derived mesenchymal stem cells [31].

In this study, we explored the behaviour of intracellular magnesium during the early phase of osteogenic differentiation in a 3D cell model built to mimic osteogenesis, and we checked the presence and measured the amount of magnesium in the mineral depositions of phenotype committed cells.

The 3D cell model was built employing osteoblast-like SaOS2 cells grown on board of collagen scaffolds and treated with an osteogenic cocktail. These cells have a mature osteoblast phenotype with high levels of ALP activity since the master regulator of osteoblast gene expression *RUNX2* is constitutively expressed in this cell line [32]. Nevertheless, osteosarcoma cells SaOS2 have been reported as able to form a calcified matrix typical of woven bone [33], thus representing a suitable in vitro tool to study the differentiation of human osteoblasts towards osteocytes.

A deeper knowledge of recruited molecular pathways has highlighted that osteosarcoma occurs when osteoblast differentiation is disrupted, e.g., by mutations, and the cells maintain different stages of primitive osteoblasts. Indeed, a new strategy of anticancer therapy is to induce the differentiation of cancer cells towards a less aggressive phenotype [34,35].

In our experiments, we observed a regular cell growth pattern in 3D scaffold and cell cycle analysis indicates that collagen matrix is appropriate for tissue-like growth of SaOS2 cells. The gene expression analysis of osteogenic markers suggests that the osteogenic commitment of cells is already well active few days after the induction. The constitutively elevated levels of *RUNX2* mRNA in control cultures is likely to account for a lack of its further upregulation when the cells have been stimulated with osteogenic cocktail [36]. On the other hand, a significant increase of *BGLAP*, and *SPP1*, together with an apparent, although non statistically significant, trend towards the upregulation of *COL1A1*, was recorded in response to treatment at both days 3 and 7, demonstrating a successful activity of the administered molecular cocktail over SaOS2 osteogenic function. The above described osteogenic marker gene profile was consistent with matrix mineralization, evaluated as calcium deposition at Alizarin Red staining, demonstrating that SaOS2 cells are a useful model for studying human osteocyte differentiation and responsiveness in vitro. Indeed, alizarin staining showed many mineral nodular aggregates in treated cells, already after three days, with red spots larger and stained more intensively at day 7. Furthermore, we characterized the depositions observed in slices by Alizarin S red staining in the whole scaffolds, exploiting the high sensitivity and spatial resolution of laboratory-based (TomoLab @ Elettra) and synchrotron-based X-ray µCT (Syrmep @ Elettra). These techniques allowed to obtain 3D spatial distribution maps of the mineral depositions generated by SaOS2 cells. We found an increase of biomineralization in differentiating SaOS2 cells revealed by near three-fold higher amount of depositions in different samples. Nevertheless, the average volume of single depositions in the scaffold with differentiating cells was lower than those of the control scaffold. We speculate that the decrease of deposition volume could be related to the evolution of the amorphous Ca compounds into crystalline HA during the biomineralization process. To corroborate this hypothesis, in a recent study on differentiating stem cells, we found that the mineral amorphous depositions rapidly evolve toward hexagonal HA crystal, similar to those present in mature human bone [37].

There are many evidences that collagen microfibrils direct the formation of nanosized HA platelets oriented parallel to the collagen fibril axis [38,39] for interfibrillar mineralization [40]. Indeed, synchrotron-based X-ray micro-tomography showed many depositions following an oriented pattern along denser filaments, as clearly evident in Figure 5. Nevertheless, we found an overexpression of *COL1A1*, the gene coding for collagen type 1, speculating that the observed filaments could be the neo-formed collagen fibrils produced by differentiating cells.

To elucidate the role of magnesium in osteoblast differentiation, we assessed its intracellular concentration by DCHQ5 fluorescent dye. A marked increase of magnesium was found, at both three and seven days from induction. In particular, we found a two-fold increase of intracellular content in the differentiating cells with respect to controls. This result is in agreement with recent publications demonstrating a pivotal role of Mg in bone differentiation involving the ERK/BMP2/Smads signaling pathway [41,42]. This finding poses the question as to whether Mg is present in the mineral Ca depositions produced in the early phase of biomineralization. Therefore, we characterized by X-ray fluorescence microscopy the chemical composition of the mineral depositions secreted by 3D cultured differentiated cells. In particular, we evaluated the Mg and P content in the mineral depositions. We found a marked co-localization of Mg with P at seven days of differentiation, indicating the presence of magnesium in the mineral depositions already in the early phases of biomineralization. As widely reported in the literature, the mineral fraction of bone is composed of HA, a non-stoichiometric Ca compound containing trace elements such as Zn, Mg which are the main substituting cations in the crystal lattice of HA [43,44].

This is the first experimental evidence on Mg presence in the mineral depositions generated in the early phase of biomineralization, suggesting that Mg incorporation in the bone forming process occurs just at the beginning of the differentiation, similarly to Zn as shown in a recent study [37].

In conclusion, in this study, we showed that there is an evident involvement of Mg in the process of cell differentiation. Moreover, an intriguing question arises: iss the increase of intracellular Mg content in differentiating cells a typical feature of the differentiation process or a peculiarity of osteosarcoma cells? To answer this question, it will be very interesting to investigate on stem cells.

## 4. Materials and Methods

### 4.1. D Cell Culture and Osteogenic Induction

A 3.8 mg/mL collagen solution pH 3-4 (RatCol®; CellSystems GmbH, Troisdorf, Germany) was used for this study. 9 parts of Rat Tail Collagen solution were mixed on ice with 1 part of formulated neutralization solution into a sterile mixing tube for a total of 10 parts for the formation of a collagen gel. 2 × 10^5^ osteosarcoma SaOS2 cells suspended in 55 µL of complete medium were added to 245 µL of collagen solution. The final Rat Tail Collagen mixture was dispensed into 96-well plates, in a final volume of 300 µL for each scaffold. These were incubated in humidified atmosphere at 37 °C with 5% CO_2_ in 1 mL of culture medium (RPMI 1640, supplemented with 10% heat inactivated FBS and 2 mM Glutamine). After 24 h, cells were treated with vehicle or osteogenic differentiation cocktail containing 20 nM 1α,25-Dihydroxyvitamin D3 (D1530, Sigma-Aldrich, Milano, Italy), 50 µM L-Ascorbic acid 2-phosphate (49752, Sigma-Aldrich, Milano, Italy) and 10 mM β-Glycerol phosphate (50020, Sigma-Aldrich, Milano, Italy), replacing the media every 48 h. At indicated time points, the scaffolds were dissolved with a 0.25% collagenase solution and then processed for following analysis. To obtain cell growth curve, viable cells from triplicate wells were counted at 1, 3, 7, and 14 days. To determine cell cycle distribution, SaOS2 cells were detached with 0.11% trypsin/0.02% EDTA, washed in PBS, and centrifuged. The pellet was resuspended in 0.01% nonidet P-40, 10 µg/mL RNase, 0.1% sodium citrate, and 50 µg/mL propidium iodide for 30 min at 37 °C in the dark. Propidium iodide fluorescence was analyzed by using a flow cytometer Bryte HS (BioRad) and cell cycle analysis was performed using the Modfit 5.0 software (Verity Software House, Topsham, ME, USA).

### 4.2. Gene Expression Analysis

Total RNA from cells collected at the indicated time points from up to three scaffolds cultured in control condition or under osteogenic induction was extracted using the NucleoSpin® RNA (Macherey Nagel, Düren, Germany) following the manufacturer’s instructions. The level of expression of the osteogenic markers runt-related transcription factor 2 (*RUNX2*), collagen type I (*COL1A1*), osteocalcin (*BGLAP*) and osteopontin (*SPP1*) was analyzed by quantitative real-time PCR (qPCR), as previously reported [45]. Briefly 500 ng of total mRNA was reverse transcribed into cDNA using the iScript™ cDNA Synthesis Kit (170-8891, Bio-Rad, Segrate, Italy) and then diluted 1:10 with water. 5 μL of cDNA was then amplified using the SsoAdvanced™ SYBR® Green Supermix (172-5261, Bio-Rad) in duplicate using the CFX Connect™ Real-Time PCR Detection System (Bio-Rad) applying a two-step protocol (2 min at 95 °C, 40 cycles 5 s at 95 °C and 30 s at 60 °C) followed by a melting step between 95 and 65 °C. Data analysis was performed with the CFX Manager™ Software (Bio-Rad) using the 2^-ΔΔC*T*^ method with Glyceraldehyde 3-phosphate dehydrogenase (GAPDH) and Hypoxanthine-guanine phosphoribosyltransferase (HPRT1) as reference genes and reporting data as fold changes over control condition. A gene study that uses an inter-run calibrator among experiments was then created to pull all the experiments together. Data are reported as mean value ± SEM of three independent biological replicates. Statistical analysis was performed using GraphPad Prism 6 software using a two-way ANOVA followed by a Sidak’s multiple comparison test. P values less than 0.05 were accepted as significant.

### 4.3. Total Intracellular Mg Quantification

After the scaffold dissolution with collagenase 0.25%, SaOS2 cells were washed twice with PBS without Ca^2+^ and Mg^2+^, counted and resuspended at 1x10^6^ cells/mL. Samples were lysed by sonifier, and 100 µL of the sample were added to 100 µL of PBS without Ca^2+^ and Mg^2+^, 22 µL of the magnesium probe DCHQ5 1.37 µM in DMSO and 1778 µL of MOPS 20 mM/Methanol 50%. Fluorescence spectra were collected with λ_ex_ 363 nm. Mg concentration was assessed comparing the fluorescence intensity at λ_em_ 510 nm of the samples with a calibration curve prepared with MgSO_4_ [19].

### 4.4. Histological Analysis

Collagen scaffolds were fixed with paraformaldehyde (4%) for 1 h at room temperature (RT), dehydrated through a graded alcohol series up to 100% and embedded in paraffin with standard methods. Histological sections (thickness, 10 μm) were cut orthogonally to the scaffold axis, deparaffinizedwith xylene, rehydrated to 70% alcohol, and stained according to standard hematoxylin/eosin staining protocol with Mayer’s hematoxylin and 1% eosin solution and mounted in permanent medium. For Alizarin red S staining, the deparaffinized rehydrated sections were rinsed rapidly in distilled water and stained with 2% Alizarin red S solution for 5 minutes. Excess dye was shaken off and stained sections were fixed with acetone (20 dips), aceton-xylene (20 dips), cleared with xylene and mounted in permanent medium [46]. Quantitative analysis of ECM mineralization was performed on images acquired from cellularized collagen scaffold slices. Control and treated data were compared. To this aim, images acquired with optical microscope were processed, using ImageJ software (National Institute of Health, USA), in order to determine the size of the red (Alizarina Red staining) area. This was calculated using the maximum entropy threshold-based image segmentation method [47]. Four separate fields out of each single microphograph were analysed.

### 4.5. X-ray Fluorescence Microscopy Analysis

The XRFM and STXM measurements were carried out at the Twinmic beamline [25] of the Elettra Synchrotron light source (Basovizza, Trieste, Italy). A Fresnel zone plate focused the incoming beam (2150 eV), monochromatized by a plane grating monochromator, to a circular spot of about 1.25 ∝m in diameter. The sample was transversely scanned in the zone plate focus, in steps of 1.2 µm. At each step the fluorescence radiation intensity (dwell time 8–10 s) was measured by eight Si-drift detectors (active area 30 mm^2^) (chips from PnDetector, Munich, Germany and electronics from XGLab, Milano, Italy) [48] concentrically mounted at a 20° grazing angle with respect to the specimen plane, at a detector-to-specimen distance of 28 mm. Simultaneously, the transmitted intensity was measured by a fast-readout electron-multiplying low-noise charge-coupled device (CCD) detector (Andor Technology, Belfast, Ireland) through an X-ray to visible light converting system [49]. Zone plate, sample, and detectors were accommodated in vacuum, thus avoiding any absorption and scattering by air. The X-ray fluorescence spectra were analysed by PyMCA software [50], which provides the total counts for the fluorescence K-lines of Mg and P.

### 4.6. Laboratory-Based X-ray Computed Microtomography Analysis

Laboratory-based X-ray computed microtomography (µCT) measurements were carried out at the TomoLab station [51] at Elettra. This instrument, based on a microfocus X-ray source (minimum focal spot size of 5 µm), can operate at a Voltage ranging between 40 and 130 kV with a maximum power of 39 W. A 12-bit, water cooled, 4008 × 2672 pixels CCD camera was used as detector. The camera chip is coupled via a fiber optic bundle to a Gadox scintillator screen and a de-magnifying optics allows to obtain an effective pixel size of 12.5 µm yielding a maximum field of view of 50 × 33 mm^2^ at sample. Due to the focal spot size of the source and the specific design of the TomoLab station [22] it was possible to perform propagation-based phase-contrast X-ray µCT measurements. The experimental parameters used for the imaged samples are summarized in Appendix A. The tomographic reconstruction was performed by the commercial software COBRA (Exxim, USA) while the 3D visualization of the reconstructed and processed data was obtained by the commercial software VGStudio MAX 2.0 (Volume Graphics, Heidelberg, Germany).

### 4.7. Synchrotron-Based X-ray Computed Microtomography Analysis

One tomographic data set for a treated sample has been acquired at the SYRMEP beamline of Elettra [24], which was operated at an electron energy of 2.0 GeV. The source of the synchrotron radiation is a bending magnet featuring a magnetic field of 1.2 T, which generates a white spectrum of synchrotron radiation with a critical energy of about 3.2 KeV. This white beam spectrum has been filtered by a 500 µm thick Be window and additional Si filter of a thickness of 500 µm resulting in a bell-shaped curve centered around 24 keV. The images were recorded in phase-contrast (propagation-based) mode. The source-to-sample distance was around 22 m while the sample-to-detector (thus the propagation) distance was set to 150 mm. The detector was a sCMOS imager (Orca Flash from Hamamatsu) optically coupled to a 45 µm thick GGG:Eu (Gd3Ga5O12:Eu) scintillator utilizing a set of optical lenses with different magnifications. sCMOS sensor comprises 2048 × 2048 pixels^2^ (with a size of 6.5 × 6.5 µm^2^) and features a dynamic range of 37,000:1. For the data set described here the highest optical magnification has been used, which translates into a field of view of 1.85 mm × 1.85 mm and a pixel size of 0.9 × 0.9 µm^2^. A total of 1800 equiangular projections have been acquired over a total scan angle of 180 degrees using an exposure time of 100 ms per projection. The tomographic reconstruction was performed utilizing an in-house software tool, the so-called SYRMEP Tomo Project (STP) software (https://github.com/ElettraSciComp/STP-Gui) [52]. Prior to image reconstruction based on a filtered back projection algorithm, each projection was independently pre-processed with a phase-retrieval algorithm based on the transport of intensity equation [53].

## Figures and Tables

**Figure 1 ijms-21-02368-f001:**
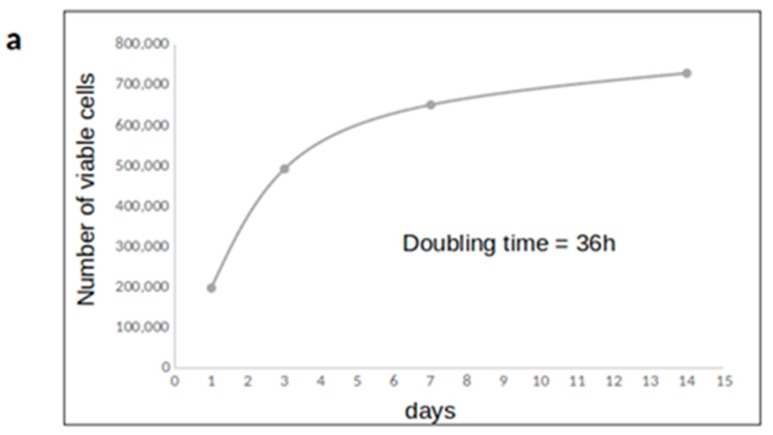
Assessment of 3D cell culture conditions. (**a**) cell growth curve; (**b**) cell cycle analysis. The figure depicts the results obtained in one experiment representative of three.

**Figure 2 ijms-21-02368-f002:**
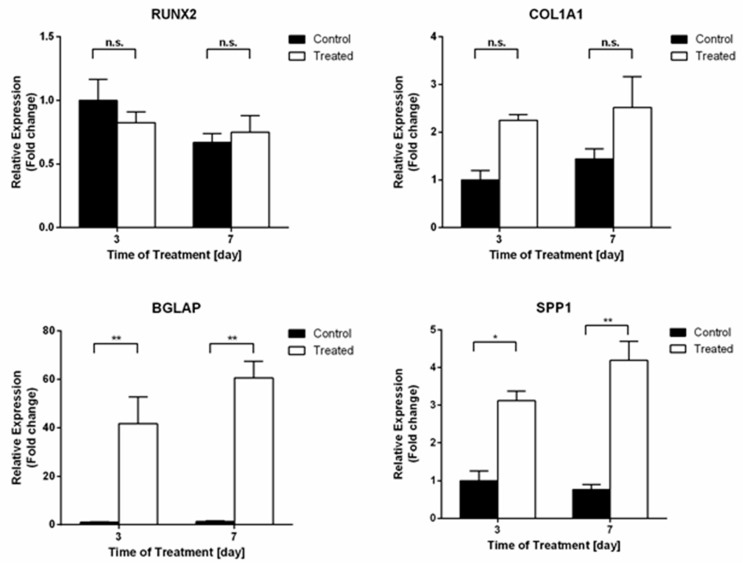
Analysis of SaOS2 cells cultured in 3D collagen scaffolds (3 and 7 days) treated with the osteogenic cocktail (20 nM 1α,25-Dihydroxyvitamin D3, 50 µM L-Ascorbic acid 2-phosphate, 10 mM β-Glycerophosphate). qPCR analysis of osteogenic markers (*RUNX2*, *COL1A1*, *BGLAP*, and *SPP1*) was performed using GAPDH and HPRT1 as reference genes (2-ΔΔC*T* method). Fold changes from control untreated cells at day 3 were calculated. Data are reported as mean ± SEM of three biological replicates. * *p* < 0.05; ** *p* < 0.01.

**Figure 3 ijms-21-02368-f003:**
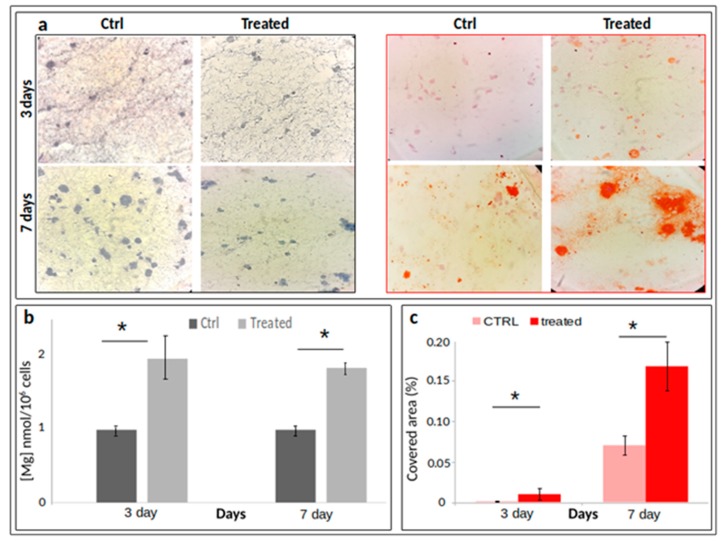
H/E ((**a**), left panel) or Alizarin Red ((**a**), right panel) staining of paraffin-embedded sections of 10 μm thickness (*n* = 3). (**b**) Fluorimetric assay of total intracellular magnesium content by DCHQ5 probe (panel b) [19]; panel (**c**) quantification of Alizarin-stained Ca depositions. Data are reported as means ± standard error of the mean, values deriving from a triplicate experiment. * *p* < 0.05.

**Figure 4 ijms-21-02368-f004:**
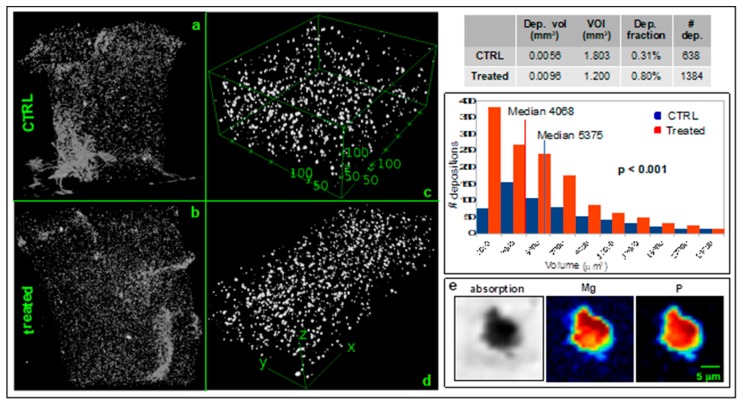
X-ray analysis of depositions released by SaOS2 cells. Panels (**a**) and (**b**) depict the 3D distribution of mineral depositions in the whole control and treated scaffolds imaged by microfocus X-ray µCT at the TomoLab station of Elettra (isotropic voxel size = 5 µm for the control and 3.3 µm for the treated sample). The dimensions of the scaffolds are approximately 80 mm^3^. Panel (**c**) and (**d**) show two selected VOIs reporting the depositions distribution in CTRL (6000 × 4300 × 700 µm^3^) and treated scaffolds (7300 × 1200 × 800 µm^3^) respectively. Table and graph show the descriptive statistical analysis of the depositions in the two representative VOIs showed in panel c (CTRL) and d (treated), respectively. The graph classified the particles dimension (µm^3^) in relation to the number of particles within the class. Panel (**e**) describes the Mg and P co-localization in a deposition released by a differentiating SaOS2 cell at the TwinMic beamline of Elettra.

**Figure 5 ijms-21-02368-f005:**
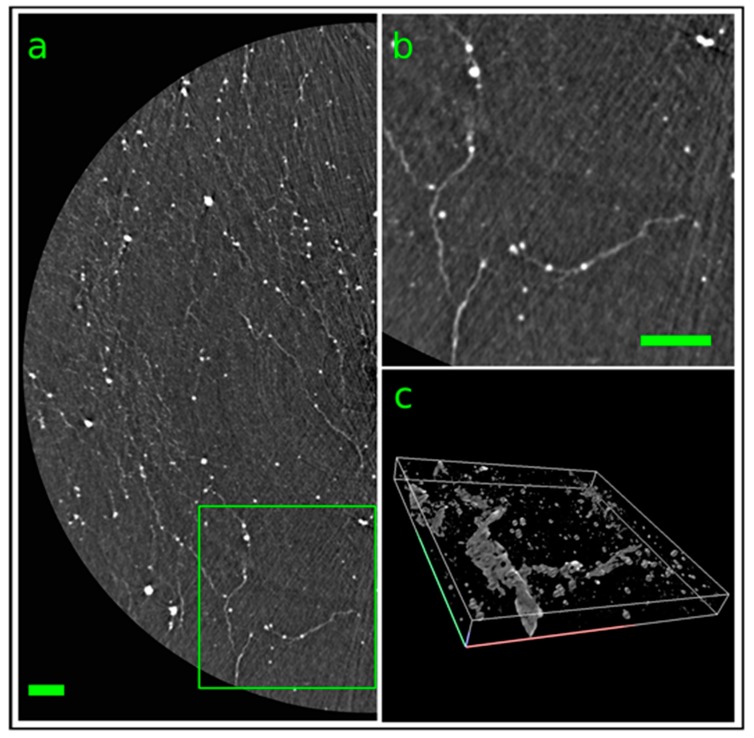
Synchrotron-based X-ray µCT analysis of one treated scaffold imaged at the SYRMEP beamline of Elettra. Panel (**a**) represents a reconstructed axial slice of the scaffold (isotropic voxel size = 0.9 µm) highlighting the presence of mineral depositions (white dots) along brighter filaments. Panel (**b**) shows a zoom of the green ROI indicated in panel (**a**). Panel (**c**) depicts a 3D rendering of 50 adjacent slices (400 × 400 × 50 µm^3^) showing the 3D distribution of the mineral depositions. Scale bars = 100 µm.

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
