# Peer review of "Analysis of Intracellular Magnesium and Mineral Depositions during Osteogenic Commitment of 3D Cultured Saos2 Cells"

_ijms, 2020, doi:10.3390/ijms21072368_

Round 1
Reviewer 1 Report
Review
Giovanna Picone and collaborators presented here original and well-written paper investigating the level of intracellular magnesium and mineral depositions during osteogenic commitment in 3d model of cultured osteosarcoma (saos-2) cells.
Presented results might be interesting and valuable in further research concerning reprogramming malignant cancer cells toward a benign phenotype.
The research presented in the article was well planned and carried out, as well as clearly described. I do not have substantive remarks; nevertheless, some minor corrections have to be made.
My first comment concerns the references. There is a lot of confusion there. The authors must check all references - one by one because the text lacks several references to the list of cited publications in the "references" chapter. In the reference list first position is missing. Square brackets and spaces are also missing (I pointed it out in yellow in the article body).
Secondly, I’m not satisfied with regards to figures. Particularly figure 1 and 3 but also 4 (in the right bottom part), includes graphs with the indistinct and illegible font. Figure 5 looks like individual photos were being shifted. Could the authors improve these figures so that they are correctly prepared and transparent?
Thirdly, in the whole text of article authors do not respect the rule of writing gene nomenclature in italics, what is a mistake and makes reading difficult.
Below is presented a list of small mistakes examples that are marked in yellow and commented in the article. Please, make appropriate corrections.
Page 1, line 35, in Keywords: there is „Osteogenesys”, it should be osteogenesis.
Page 2, line 46: there is “Cell cycle analysis confirmed that cells were highly proliferating at 3 days,…” it should be “for 3 days” or “at day 3”.
Page 5, line 9 and 14: at day 7 or for 7 days?
Page 6, line 24: Please unify spelling: TwinMic as above, in line 14.
Page 8, line 18: Hydroxyapatite with the shortcut (HA) is mentioned first time in the introduction (page 1, line 41) and there is no need to use the full name below in the text.
Page 8, line 48: a mistake in paragraph title occurred - incorrect numbering and space placing.
Page 9, line 2: Is the unit correct? It should be "μl" rather.
Page 9, line 4: Something is wrong here. Shouldn’t it be: with 5% CO2?
Moreover, in the whole article, the name of the cell line should be unified.
I recommend the source name:
https://www.lgcstandards-atcc.org/Products/All/HTB-85.aspx?geo_country=pl
I’m not a native English speaker; so I do not attempt to assess grammar correctness.

Author Response
Referee 1:
Giovanna Picone and collaborators presented here original and well-written paper investigating the level of intracellular magnesium and mineral depositions during osteogenic commitment in 3d model of cultured osteosarcoma (saos-2) cells.
Presented results might be interesting and valuable in further research concerning reprogramming malignant cancer cells toward a benign phenotype.
The research presented in the article was well planned and carried out, as well as clearly described. I do not have substantive remarks; nevertheless, some minor corrections have to be made.
My first comment concerns the references. There is a lot of confusion there. The authors must check all references - one by one because the text lacks several references to the list of cited publications in the "references" chapter. In the reference list first position is missing. Square brackets and spaces are also missing (I pointed it out in yellow in the article body).
Secondly, I’m not satisfied with regards to figures. Particularly figure 1 and 3 but also 4 (in the right bottom part), includes graphs with the indistinct and illegible font. Figure 5 looks like individual photos were being shifted. Could the authors improve these figures so that they are correctly prepared and transparent?
We apologize for the inconvenience but the confusion in both references and figures was due to the conversion of the manuscript from libreoffice to microsoft word format. In the revised manuscript, we improve the quality of figures and we check all the references. In addition, we modify figure 5 separating the three panels.
Thirdly, in the whole text of article authors do not respect the rule of writing gene nomenclature in italics, what is a mistake and makes reading difficult.
We modified the text accordingly
Below is presented a list of small mistakes examples that are marked in yellow and commented in the article. Please, make appropriate corrections.
Page 1, line 35, in Keywords: there is Osteogenesys”, it should be osteogenesis. We signalled the typos to the editorial office as they can change the keyword
Page 2, line 46: there is “Cell cycle analysis confirmed that cells were highly proliferating at 3 days,…” it should be “for 3 days” or “at day 3”. Modified
Page 5, line 9 and 14: at day 7 or for 7 days? Modified
Page 6, line 24: Please unify spelling: TwinMic as above, in line 14. Modified
Page 8, line 18: Hydroxyapatite with the shortcut (HA) is mentioned first time in the introduction (page 1, line 41) and there is no need to use the full name below in the text. Modified
Page 8, line 48: a mistake in paragraph title occurred - incorrect numbering and space placing. Modified
Page 9, line 2: Is the unit correct? It should be "μl" rather. Modified
Page 9, line 4: Something is wrong here. Shouldn’t it be: with 5% CO2? Modified
Moreover, in the whole article, the name of the cell line should be unified.
I recommend the source name: https://www.lgcstandards-atcc.org/Products/All/HTB-85.aspx?geo_country=pl Modified
I’m not a native English speaker; so I do not attempt to assess grammar correctness.
Reviewer 2 Report
Major critique:
The main problem of this study is that the presented data are very descriptive and do not provide a conceptual advance of the field. Specifically, it is not clear as to how exactly a 3D cell culture of SaOS cells may help uncover basic mechanisms of osteogenesis, the cellular balance of Mg2+ and the formation of extracellular mineral deposits. Furthermore, the authors did not show evidence for a direct causal link between intracellular Mg2+ levels and osteogenesis or deposition of extracellular minerals. Taken together, the results remain very descriptive and do not allow for any meaningful mechanistic conclusions.
Specific points:
Introduction: The notion that about 50% of body Mg is stored in bones needs references containing the original experimental data. How exactly is Mg2+ distributed in bone?
Results:
Fig. 1: The overall design of the experiment is not clear. What are the biological and technical replicates of the study? Also, a corresponding statistical analysis should be performed and reported.
Fig. 2: Apparently, changes in the expression levels of RUNX2 and COL1A1 were not statistically significant. This should be indicated clearly in the figure. If RUNX2 and COL1A1 are not changed, the conclusion that SaOS cells developed towards osteogenic commitment relies on up-regulation of two other markers (BGLAP and SPP). Hence, more comprehensive evidence should be presented to support such a conclusion.
Fig. 3A: Experiments with Alizarin Red have to be quantified and subjected to statistical analysis.
Fig. 4: The resolution of this figure is very poor (especially the labelling of the bar graph) likely due to the conversion of the original figure to a PDF. Accordingly, the reviewer is not able to interpret and assess the corresponding data.
Fig. 5: The reviewer is not able to interpret the images presented in this figure (apparently, it is a problem of PDF creation). The corresponding figure legend and text are not very helpful to understand the data presented. Again, these observations have to be quantified with a clear description of biological and technical replicates and statistical analysis.
Author Response
Referee 2
The main problem of this study is that the presented data are very descriptive and do not provide a conceptual advance of the field. Specifically, it is not clear as to how exactly a 3D cell culture of SaOS cells may help uncover basic mechanisms of osteogenesis, the cellular balance of Mg2+ and the formation of extracellular mineral deposits. Furthermore, the authors did not show evidence for a direct causal link between intracellular Mg2+ levels and osteogenesis or deposition of extracellular minerals. Taken together, the results remain very descriptive and do not allow for any meaningful mechanistic conclusions.
The aim of this work is the study of the involvement of Mg in osteoblastic differentiation and not the 3D cell culture model. To clarify this we changed the introduction accordingly. In this work we used a 3D model in order to better mimic the physiological conditions of bone tissue, as well documented in literature. Indeed, finding evidence for a direct causal link between intracellular Mg2+ levels and osteogenesis or deposition of extracellular minerals, is our final goal such as for a good part of the scientific community working on Mg. We are actively engaged to give our contribution to solve a quite complex puzzle on which several research groups are committed.
Specific points:
Introduction: The notion that about 50% of body Mg is stored in bones needs references containing the original experimental data. How exactly is Mg2+ distributed in bone?
We have added the reference reporting the notion that about 50% of body Mg is stored in bones. In addition we have specified that magnesium is mainly distributed in hydroxyapatite crystals, where it is understood to form a fixed and dynamic pool.
[Effects of Magnesium on Mechanical Properties of Human Bone Raviraj Havaldar, S. C. Pilli, B. B. Putti Materials Science 2013 DOI:10.9790/3008-0730814]
Fig. 1: The overall design of the experiment is not clear. What are the biological and technical replicates of the study? Also, a corresponding statistical analysis should be performed and reported.
The overall design of the experiment described in fig1 is to assess the optimal cell growth condition on the 3D collagen scaffold. We monitored: the cells viability up to 14 days and the cell cycle progression at 3 and 7 days. Figure 1 depicts the results obtained in one experiment representative of three biological replicates. We thank the referee and we modified the manuscript accordingly.
Fig. 2: Apparently, changes in the expression levels of RUNX2 and COL1A1 were not statistically significant. This should be indicated clearly in the figure.
Absence of statistical significance of changes in the expression levels of RUNX2 and COL1A1 are now clearly indicated in figure 2.
If RUNX2 and COL1A1 are not changed, the conclusion that SaOS cells developed towards osteogenic commitment relies on up-regulation of two other markers (BGLAP and SPP). Hence, more comprehensive evidence should be presented to support such a conclusion.
We fully understand the reviewer’s concern and in this respect the abstract, introduction and discussion sections of the manuscript have been revised describing as “bone phenotype modulation” the up-regulation of BGLAP and SPP osteogenic markers in SaOS2 cells. BGLAP codes for osteocalcin, the most abundant bone-specific non-collagenous protein synthesized by osteoblasts, that serves as a marker to evaluate osteogenic maturation and bone formation [e.g. see Ducy P, Desbois C, Boyce B, et al. Increased bone formation in osteocalcin-deficient mice. Nature. 1996;382(6590):448–452]. SPP codes for osteopontin, a structural protein synthesized by preosteoblasts, osteoblasts, and osteocytes. [e.g. see Butler WT. The nature and significance of osteopontin. Connect. Tissue Res. 1989;23(2–3):123–136]. Together with the istochemical assessment of matrix mineralizazion (using Alizarin red) both are accepted indexes to monitor the osteogenic phenotype in human osteoblast-like SaOS2 cells, considered in the scientific literature as a representative model of human bone-forming cells [e.g. see Saldaña L. In search of representative models of human bone-forming cells for cytocompatibility studies. Acta Biomaterialia 2011;7:4210–4221 and/or Czekanska EM. In search of an osteoblast cell model for in vitro research 2012;24:1-17).
Fig. 3A: Experiments with Alizarin Red have to be quantified and subjected to statistical analysis.
Thank to the referee for this suggestion. We modify the manuscript adding an histogram on the Alizarin red quantification in figure 3 panel (c).
Fig. 4: The resolution of this figure is very poor (especially the labelling of the bar graph) likely due to the conversion of the original figure to a PDF. Accordingly, the reviewer is not able to interpret and assess the corresponding data.
We apologize for the inconvenience, the poor quality of the figures derived from the conversion of the manuscript in docx format from the native libreoffice. We improved quality of figure 4
Fig. 5: The reviewer is not able to interpret the images presented in this figure (apparently, it is a problem of PDF creation). The corresponding figure legend and text are not very helpful to understand the data presented. Again, these observations have to be quantified with a clear description of biological and technical replicates and statistical analysis.
Figure 5 depicts a single measurements of a treated collagen scaffold acquired by synchrotron-based X-ray μCT. This picture provides the proof of principle of the use of synchrotron X-ray μCT for the study of biomineralization in 3D soft-matter collagen scaffold at the micrometer scale. We follow the suggestion of the referee and we modify the figure dividing the figure in three separated panels and we improve the caption to make it more clear for the readers.
Round 2
Reviewer 2 Report
The authors have addressed most of the issues raised and the manuscript has been improved.